

# Trophic niches of a seabird assemblage in Bass Strait, south-eastern Australia

Aymeric Fromant[1,2], Nicole Schumann[1], Peter Dann[3], Yves Cherel[2] and John P.Y. Arnould[1]

[1] School of Life and Environmental Sciences, Deakin University, Burwood, VIC, Australia
[2] Centre d'Edutes Biologiques de Chizé (CEBC), UMR 7372 du CNRS—La Rochelle Université, Villiers-en-bois, France
[3] Research Department, Phillip Island Nature Parks, Cowes, VIC, Australia

## ABSTRACT

The foraging niches of seabirds are driven by a variety of factors, including competition for prey that promotes divergence in trophic niches. Bass Strait, south-eastern Australia, is a key region for seabirds, with little penguins *Eudyptula minor*, short-tailed shearwaters *Ardenna tenuirostris*, fairy prions *Pachyptila turtur* and common diving-petrels *Pelecanoides urinatrix* being particularly abundant in the region. The trophic niches of these species were investigated using isotopic values in whole blood and by identifying prey remains in stomach contents. The four species occupied different isotopic niches that varied among years, seasons and regions. Little penguins consumed mainly fish whereas the three procellariforms primarily consumed coastal krill *Nyctiphanes australis*. The dietary similarities between the procellariforms suggest that food resources are segregated in other ways, with interspecific differences in isotope niches possibly reflecting differential consumption of key prey, divergent foraging locations and depth, and differences in breeding phenology. Because oceanographic changes predicted to occur due to climate change may result in reduced coastal krill availability, adversely affecting these seabird predators, further information on foraging zones and feeding behaviour of small procellariform species is needed to elucidate more fully the segregation of foraging niches, the capacity of seabirds to adapt to climate change and the potential for interspecific competition in the region.

## INTRODUCTION

Seabirds are major consumers of marine biomass, feeding on a variety of fish, cephalopods and crustaceans (*Ridoux, 1994*; *Brooke, 2004*). The foraging niche of seabirds is influenced by a range of factors, including environmental conditions (*Waugh & Weimerskirch, 2003*; *Amélineau et al., 2016*; *Jakubas et al., 2017*), prey availability (*Baird, 1991*; *Camprasse et al., 2017*), morphological characteristics and their influence on flight performance (*Phillips et al., 2004*; *Phillips, Silk & Croxall, 2005*; *Navarro et al., 2013*), and interspecific competition (*González-Solís, Croxall & Afanasyev, 2008*; *Phillips et al., 2008*; *Kokubun et al., 2016*). Competition is thought to promote foraging niche divergence since species occupying the same ecological niche cannot theoretically coexist through time (*Schoener,*

Corresponding author
Aymeric Fromant,
afromant@deakin.edu.au

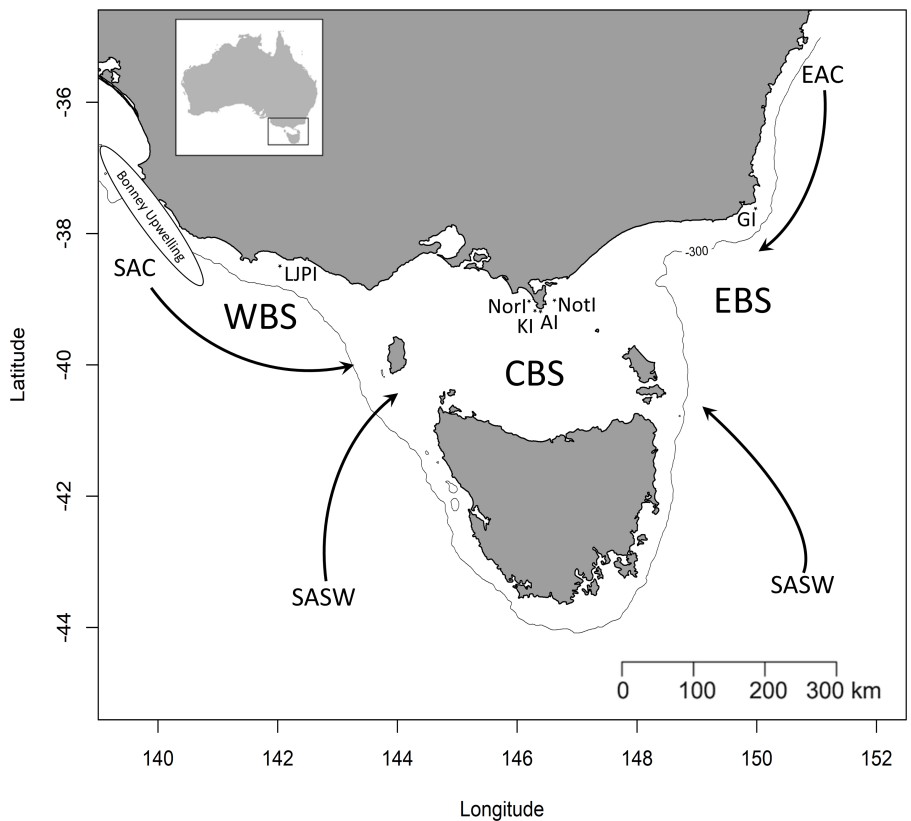

**Figure 1** **Simplified representation of the three study areas and the major water masses influencing the region.** Western Bass Strait (WBS); Central Bass Strait (CBS); Eastern Bass Strait (EBS); Lady Julia Percy Island (LJPI); Norman Island (NorI); Kanowna Island (KI); Anser Island (AI); Notch Island (NotI); Gabo Island (GI); South Australian Current (SAC); Sub-Antarctic Surface Water (SASW); East Australian Current (EAC) from *Sandery & Kämpf, 2007*. The solid line indicates the location of the 300 m isobath.

*1974*). Seabirds may separate their resources on several dimensions, with studies showing divergence in foraging zone (*González-Solís, Croxall & Afanasyev, 2008*; *Barger et al., 2016*), diving depth (*Mori & Boyd, 2004*), the timing of breeding (*Granroth-Wilding & Phillips, 2019*), and seasonal patterns of activity (*Phillips et al., 2008*). In particular, divergence in diet has been proposed as an important mechanism in reducing niche overlap (*Ridoux, 1994*; *Surman & Wooller, 2003*; *Pratte, Robertson & Mallory, 2017*).

Bass Strait, the shallow continental shelf area between mainland Australia and Tasmania (Fig. 1), is a key region for Australian seabirds, supporting a large proportion of breeding populations of at least 11 major species (Table 1; *Ross et al., 2001*). This area is considered a region of low primary productivity (*Gibbs, Jr & Longmore, 1986*; *Gibbs et al., 1991*) that occurs at the confluence of three main ocean currents. The warm, oligotrophic waters of the East Australian Current (EAC) flow southward along the eastern edge of Bass Strait (*Ridgeway & Godfrey, 1997*; *Sandery & Kämpf, 2007*) while the South Australian Current (SAC) advects warm water from the west onto the shelf which then flows eastward through Bass Strait (*Sandery & Kämpf, 2007*). The latter is the major source of Bass Strait water and
**Table 1 Main species of seabirds breeding in Bass Strait, indicating the estimated number of breeding pairs, their proportion of the total Australian populations (based on *Ross et al., 2001*), and the major groups of prey consumed.** Some of the population estimates were not updated for at least three decades (e.g., *Brothers et al., 2001*) and may represent a source of error.

| Species | Abundance (number of breeding pairs) | % of the Australian population | Groups of main prey | Reference |
|---|---|---|---|---|
| Shy albatross | 5,200 | 35% | Fish/cephalopods | *Alderman et al. (2011)*, *Hedd & Gales (2001)* |
| Short-tailed shearwater[a] | 14,600,000 | 75% | Crustaceans/Fish | *Weimerskirch & Cherel (1998)*, *Brothers et al. (2001)*, *Schumann, Dann & Arnould (2014)* |
| Common diving petrel[a] | 98,500 | 63% | Crustaceans | *Brothers et al. (2001)*, *Schumann, Arnould & Dann (2008)*, *Schumann, Dann & Arnould (2014)* |
| Fairy prion[a] | 97,000 | 7% | Crustaceans | *Brothers et al. (2001)*, *Schumann, Dann & Arnould (2014)* |
| White-faced storm petrel | 94,500 | 25% | Crustaceans | *Brothers et al. (2001)*, *Underwood (2012)* |
| Little penguin[a] | 353,000 | 82% | Fish | *Cullen, Montague & Hull (1992)*, *Brothers et al. (2001)*, *Dann & Norman (2006)*, *Schumann, Dann & Arnould (2014)* |
| Australasian gannet | 16,800 | 85% | Fish | *Bunce et al. (2002)*, *Bunce (2001)* |
| Black faced cormorant | 4,400 | 55% | Fish | *Brothers et al. (2001)*, *Taylor, Dann & Arnould (2013)* |
| Pacific gull | 1,500 | 82% | Scavenge - polyvorous | *Brothers et al. (2001)*, *Leitch, Dann & Arnould (2014)* |
| Silver gull | 50,000 | 35% | Scavenge - polyvorous | *Brothers et al. (2001)*, *Leitch, Dann & Arnould (2014)* |
| Crested tern | 10,400 | 13% | Fish | *Brothers et al. (2001)*, *Chiaradia et al. (2012)* |

Notes.
[a]Study species.

is strongest in winter (*Ridgeway & Condie, 2004*; *Sandery & Kämpf, 2007*). In summer, a weakening or reversal of this eastward-flowing trend occurs (*Gibbs, Jr & Longmore, 1986*; *Sandery & Kämpf, 2007*). Finally, in winter, cold, nutrient-rich sub-Antarctic surface water (SASW) enters Bass Strait from the south (*Gibbs, 1992*) where it mixes with the EAC and SAC along the sub-Tropical Convergence (STC, *Prince, 2001*).

The relative influence of the currents and upwelling systems affecting Bass Strait varies spatially, seasonally and inter-annually (*Prince, 2001*; *Sandery & Kämpf, 2005*). This affects the reproductive success of seabirds in Bass Strait, presumably due to shifts in prey availability (*Mickelson, Dann & Cullen, 1992*). Additionally, climate change is predicted to weaken the SAC (*Feng, Caputi & Pearce, 2012*) and increase the strength of the EAC, resulting in warming along the path of its strengthening (*Cai et al., 2005*). This is likely to have a considerable impact on the marine ecosystem of Bass Strait. Seabird assemblages in other parts of the world have shown differential responses to shifts in ocean regimes in parameters such as breeding success, population size and survivorship due, at least in part, to changes in prey availability (*Croxall, Trathan & Murphy, 2002*). Knowledge of the trophic relationships and diets of Bass Strait seabirds is crucial for predicting their population responses to environmental change. This information is important for the conservation of these marine predators and for the refinement of sustainable fisheries management practices. At present, the trophic structure of Bass Strait seabird community is poorly understood, with trophic niches of most pelagic species not yet described or based on a few localised studies that did not address spatial or temporal variation. Accordingly, it is not known whether, or how, they diverge in foraging niche.
The little penguin *Eudyptula minor*, short-tailed shearwater *Ardenna tenuirostris*, fairy prion *Pachyptila turtur* and common diving petrel *Pelecanoides urinatrix* are the most abundant and ubiquitous seabirds in Bass Strait, breeding sympatrically on numerous offshore islands (*Schumann, Dann & Arnould, 2014*). They are known to feed on a variety of fish, cephalopod and/or crustacean prey (*Harper, 1976*; *Montague, Cullen & Fitzherbert, 1986*; *Schumann, Arnould & Dann, 2008*; *Chiaradia et al., 2010*). Previous foraging ecology studies indicate that the little penguin is an inshore forager relying mainly on small pelagic schooling fish (*Cullen, Montague & Hull, 1992*; *Chiaradia et al., 2010*), while the pelagic short-tailed shearwater, with a foraging range extending to the Antarctic waters (*Woehler, Raymond & Watts, 2006*; *Cleeland, Lea & Hindell, 2014*), and the smaller and more neritic fairy prion and common diving petrel, feed on a wide range of small prey, concentrating predominantly on coastal krill (*Nyctiphanes australis*) and myctophid fish (*Harper, 1976*; *Weimerskirch & Cherel, 1998*; *Schumann, Arnould & Dann, 2008*). However, in Bass Strait, information on spatial and temporal variation in the ecology of these species is limited to the at-sea foraging behaviour of little penguins and short-tailed shearwaters (e.g., *Collins, Cullen & Dann, 1999*; *Chiaradia et al., 2010*; *Berlincourt & Arnould, 2015a*; *Berlincourt & Arnould, 2015b*) and there is almost no information on the small procellariforms (*Underwood, 2012*).

The aims of the present study, therefore, were to: (1) determine the trophic niche of the study seabirds using both stomach contents and stable isotope analysis; (2) investigate temporal (inter-annual and seasonal) and geographic variation in their isotopic niche; and (3) assess the degree of niche segregation between these four abundant species within Bass Strait.

## MATERIAL & METHODS

The study was conducted during the winters (July–August) of 2008–2010 and summers (January–February) of 2009–2011 in three regions of Bass Strait, south-eastern Australia (Fig. 1). In summer, the study species were sampled in mid, early and late chick-rearing, respectively for the little penguin (*Reilly & Cullen, 1981*), the short-tailed shearwater (*Vertigan, 2010*) and the fairy prion (*Harper, 1976*) (Fig. 2). In winter, sampling occurred during the inter-breeding period for the little penguin (*Reilly & Cullen, 1981*) and fairy prion (*Harper, 1976*) and incubation for the common diving petrel (*Schumann, Arnould & Dann, 2008*) (Fig. 2). The trophic niches of little penguins, short-tailed shearwaters, fairy prions and common diving petrels were determined using two complementary techniques. Trophic information was derived from stable isotope values in whole blood of each species in western, central and eastern Bass Strait, and stomach samples were collected from the seabirds in central Bass Strait to assess the relative importance of prey and inform interpretation of the stable isotope results. Procellariiform study species were banded and little penguins were micro-chipped to avoid sampling the same individual more than once. All research was conducted under permit from Deakin University (animal ethic permit: AWC A9-2008) and the Department of Sustainability and Environment (Permit No. 10004530), and access to the islands was provided by Parks Victoria.

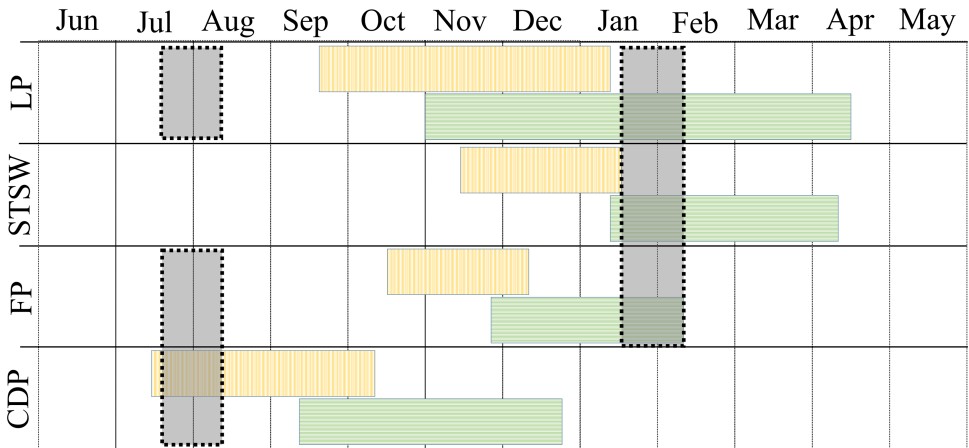

**Figure 2  Phenology and sampling period of little penguin (LP), short-tailed shearwater (STSW), fairy prion (FP) and common diving petrel (CDP) in Bass Strait.** Blocks with vertical and horizontal lines correspond to incubation and chick-rearing period, respectively. Grey shaded blocks correspond to the winter and summer sampling periods.

## Dietary analysis

Stomach contents analysis provides information on the composition and abundance of prey consumed (*Duffy & Jackson, 1986*). Stomach samples were collected from the four seabird species on Notch (38°56′S, 146°37′E) and Kanowna (39°10′S, 148°16′E) Islands in central Bass Strait (Fig. 1). Adult little penguins were sampled in both winter and summer ($n = 20$ and 22, respectively), short-tailed shearwater and fairy prion diet samples were collected in summer only ($n = 51$ and 20, respectively) and common diving petrels were sampled in winter only ($n = 6$) (Fig. 2). Due to logistical constraints and few individuals onshore in some years, it was not possible to sample all species in all years.

Adult birds were captured as they came ashore at night after foraging at sea. Diet samples were collected using the water-offloading technique. While it is possible that not all stomach contents were retrieved, stomach flushing is an effective technique for diet estimation in seabirds (*Gales, 1987*) and, in most cases, the majority of prey remains were ejected on the second (final) flush, as evidenced by clear water being ejected. After flushing, birds were given an electrolyte solution of Vytrate or Lectade (Jurox Pty Ltd, NSW), and placed into boxes for recovery (*Chiaradia, Costalunga & Kerry, 2003*). Little penguins and short-tailed shearwaters were also provided with a meal, delivered via a stomach tube, of homogenised pilchard (purchased snap-frozen and unsalted) immediately before release.

Diet samples were frozen ($-20\,^{\circ}\mathrm{C}$) or stored in 70% ethanol after collection. Crustacean remains were initially identified to lower taxonomic levels with reference to *Ritz et al. (2003)*. Sagittal otoliths, scales, fish mouth parts and cephalopod beaks were identified by comparison to reference atlases (*Neira, Miskiewicz & Trnski, 1998*; *Lu & Ickeringill, 2002*; *Furlani, Gales & Pemberton, 2007*) and collections (held by Phillip Island Nature Parks and Deakin University).

Fresh prey items were washed with water and separated from accumulated ones. To estimate the numerical abundance of crustacean prey, the heads of amphipods and whole bodies of copepods, isopods and crab megalopa were counted directly while individual eyes of krill and stomatopods were counted and divided by two. Left and right otoliths were counted and the side comprising the greater number considered representative of the minimum number of each fish taxon per sample. Where otoliths were unidentifiable, their abundance was halved and rounded to the nearest number. Similarly, the highest number of upper or lower squid beaks in a sample was used to estimate the abundance of cephalopods and only unbroken beaks were measured to estimate size (*Tollit et al., 1997*). Hard prey remains that could not be quantified, such as fish scales, were assigned a numerical abundance of one. The frequency of occurrence of prey remains was calculated as the proportion of samples containing identifiable remains in which a particular prey type occurred while the numerical abundance was expressed as the mean number of each prey taxon encountered in samples.

## Stable isotope analyses

Stable carbon ($\delta^{13}$C) and nitrogen ($\delta^{15}$N) values in tissues have been used to infer the diet of a range of marine species (*Hobson & Welch, 1992*; *Hobson, 1993*; *Cherel & Hobson, 2007*). Stable isotope values of $\delta^{13}$C values allow discrimination between benthic and pelagic prey (e.g., *Cherel & Hobson, 2007*) and inshore and offshore feeding (*Hobson, Piatt & Pitocchel, 1994b*), while $\delta^{15}$N in tissues show enrichment with increasing trophic levels (*Hobson, Piatt & Pitocchel, 1994a*). Information derived from whole blood, as used in the present study, reflects dietary integration of approximately four weeks (*Bearhop et al., 2002*).

Blood samples (<0.2 ml) were collected from seabirds resident in western (WBS - Lady Julia Percy Island: 38°25′S, 142°00′E), central (CBS - Notch, Kanowna, Norman: 39°02′S, 146°12′E and Anser Islands: 39°09′S, 146°18′E) and eastern Bass Strait (EBS - Gabo Island: 37°34′S, 149°55′E). A total of 278 (167 in summer, 111 in winter), 177 (summer only), 88 (66 in summer, 22 in winter) and 38 (winter only) stable isotope profiles were obtained from little penguins, short-tailed shearwaters, fairy prions and common diving petrels, respectively. Adult individuals (only birds that were not sampled for diet determination) were captured as they returned to their nesting burrows at night or taken from their burrows during the day. Little penguins and short-tailed shearwaters were captured by hand, common diving-petrels were captured in mist nets, and fairy prions were captured by hand or using hand nets as they approached their burrows. Upon capture, blood was collected into a heparinised syringe via venipuncture of the tarsal vein or an inter-digital vein in the foot.

Blood samples were stored frozen (−20 °C) and, prior to analysis, oven dried (60 °C) and homogenised using a mortar and pestle. The low lipid content of whole blood does not typically necessitate lipid extraction (*Cherel, Hobson & Hassani, 2005*). Indeed, all mean values of C:N mass ratio encompassed a narrow range (3.1–3.7) indicating low lipid content and, thus, allowing accurate comparisons of $\delta^{13}$C values among groups (*Bond & Jones, 2009*). Isotope ratios in whole blood were measured using a continuous-flow isotope ratio mass spectrometer, with analyses conducted by the Isotope Ratio Mass Spectrometry

service in the Research School of Biology, Australian National University (Canberra, Australia). The values of stable isotope abundances were expressed in $\delta$-notation as the deviation from standards in parts per thousand according to the equation:

$$\delta X = [(R_{sample}/R_{standard})-1]$$

where $X$ is $^{15}$N or $^{13}$C and $R$ represents the corresponding $^{15}$N/$^{14}$N or $^{13}$C/$^{12}$C ratios (*Hobson, Piatt & Pitocchel, 1994a*). $R_{standard}$ values were based on Vienna Pee Dee Belemnite for $^{13}$C, and atmospheric nitrogen (N$^2$) for $^{15}$N. Based on variation between repeats of a standard material, measurement error was estimated to be $\pm 0.20$ and $\pm 0.15$‰ for $\delta^{15}$N and $\delta^{13}$C, respectively.

## Statistical analyses

All statistical analyses were conducted in the R statistical environment 3.5.1 (*R Development Core Team, 2018*). To investigate the effect of geographic, inter-annual and seasonal variations in stable isotope values, generalised linear models (GLM) were fitted using the *lme4* package (*Bates et al., 2014*). Terms were added sequentially, model selection was based on Akaike's information criterion (AIC), and the global models were checked to ensure normality and homoscedasticity of the residuals. Post-hoc tests were conducted using analyses of variance (ANOVA) and *t*-tests, or Kruskal-Wallis and Wilcoxon rank sum tests where transformations did not improve data distributions. The stable isotope Bayesian ellipses in R (*SIBER* package; *Jackson et al., 2011*) were used to determine the isotopic niche width of each species in each region, for each year and season. The Standard Ellipse Area corrected (SEA$_C$; 40% probability of containing a subsequently sampled datum regardless of sample size) was used to quantify niche width. The Bayesian estimate of the standard ellipse and its area (SEA$_B$) were used to measure the overlap of the isotopic niches between groups (*Jackson et al., 2011*). The niche overlap was estimated as the isotopic area of overlap from the maximum likelihood fitted ellipses of two given groups.

## RESULTS

### Diet

Stomach content samples were obtained from individuals between August 2008 and January 2011 in order to inform interpretation of the stable isotope results. Samples were collected from little penguins in both winter and summer (2008–2009), from short-tailed shearwaters in summer 2009 and 2010, from fairy prions in summer 2011 and from common diving petrels in winter 2008 and 2009. Overall, 79, 84, 95 and 40% of little penguin, short-tailed shearwater, fairy prion and common diving-petrel samples, respectively, contained identifiable fresh prey remains.

Stomach samples of studied seabird species contained remains of fish, cephalopods and crustaceans (Table 2). Not all taxa could be identified to species level. Little penguins ingested crustaceans, comprising isopods, amphipods and/or copepods, but consumed mainly jack mackerel in winter and Australian anchovy *Engraulis australis* in summer, though high numbers of post-larval fish were also ingested in summer (Table S1). The diets of all three procellariform species were dominated by euphausiids, particularly coastal krill (*Nyctiphanes australis*), representing 78–96% of the mean number of prey items consumed
**Table 2** Percentage of numerical abundance of the main groups of prey recovered from stomach contents of little penguins, short-tailed shearwaters, fairy prions and common diving petrels in Central Bass Strait.

|  | Little penguin ($n = 42$) | Short-tailed shearwater ($n = 51$) | Fairy prion ($n = 20$) | Common diving petrel ($n = 6$) |
|---|---|---|---|---|
| Fish (%) | 74.1 | 1.8 | >0.1 | – |
| Cephalopods (%) | 4.8 | 0.2 | – | – |
| Crustaceans (%) | 21.1 | 98.0 | 99.9 | 100 |

by these species (Table S2 and S3). Other important prey taxa included *Euphausia* sp. and the hyperiid amphipod *Themisto australis* for short-tailed shearwaters, the megalopa stage of a swimming crab species *Ovalipes* sp. for fairy prions and hyperiid amphipods for common diving-petrels. For short-tailed shearwaters, the abundance of crustaceans was higher in 2010 than in 2009 (Wilcoxon-test, $w = 126.5$, $p$-value $< 0.01$), mainly driven by the variation in number per samples of coastal krill ($n = 118.9 \pm 52.6$ and $n = 969.6 \pm 194.2$ in 2009 and 2010, respectively).

## Stable isotopes analysis

Blood samples were collected from all four species in WBS and CBS, and from little penguins and short-tailed shearwaters in EBS (Tables 3 and 4). Values of $\delta^{13}$C ranged between $-20.6$ and $-18.1$ ‰ for little penguins, between $-23.7$ and $-20.4$ ‰ for short-tailed shearwaters, between $-21.0$ and $-18.3$ ‰ for fairy prions and between $-21.4$ and $-19.4$ ‰ for common diving petrels. Whole blood $\delta^{15}$N values ranged between 10.8 and 16.0 ‰ for little penguins, between 7.8 and 11.4 ‰ for short-tailed shearwaters, between 8.8 and 14.8 ‰ for fairy prions and between 10.9 and 14.5 ‰ for common diving petrels (Fig. S1).

For all four species, stable isotope values in whole blood showed intraspecific variation between regions and years (Figs. 3 and 4). Inter-annual variations of $\delta^{13}$C values were significant in all species in most regions (Paired $t$-test or Wilcoxon-test: $P < 0.01$) except for short-tailed shearwater and common diving petrel in CBS ($t$-test: $P > 0.07$). While there was no pattern in $\delta^{13}$C differences between regions for the short-tailed shearwater, for the three other species values in CBS were generally lower than those from WBS (0.56 ‰ to 1.40 ‰ lower) (Tables 3 and 4). Indeed, for the little penguin, fairy prion and common diving petrel, the variable "region" explained, respectively, 43, 37 and 58% of the variance for the $\delta^{13}$C model, but only 2% for the short-tailed shearwater. For $\delta^{15}$N, the best models retained, with interactions, the $\delta^{13}$C, the region and the year (and season for the little penguin and fairy prion) as significant variables explaining 60% to 93% of the deviance (Table S4). While "season" explained 27.1% of the model for the fairy prion (winter data available only for WBS in 2009), this variable was not significant for the little penguin (accounting for only 0.2% of the variation). No inter-seasonal variations of $\delta^{13}$C in blood of little penguin and fairy prion were found (except in CBS, paired $t$-test or Wilcoxon-test: $P < 0.01$). Significant inter-annual differences were recorded in all regions (Paired $t$-test or Wilcoxon-test: $P < 0.01$), but no clear pattern was detected in the values or the isotopic niche metrics (Tables 3 and 4 and Table S5). Similarly, for each year, the $\delta^{15}$N values varied

Fromant et al. (2020), *PeerJ*, DOI 10.7717/peerj.8700

**Table 3  Summer mean (±SD) $\delta^{13}$C and $\delta^{15}$N values (‰) in whole blood of little penguins, short-tailed shearwaters and fairy prions from western, central and eastern Bass Strait.**  The samples were collected in summer (January–February) 2009, 2010 and 2011.

| | | Western Bass Strait | | | Central Bass Strait | | | Eastern Bass Strait | | |
|---|---|---|---|---|---|---|---|---|---|---|
| | | Little penguin | Short-tailed shearwater | Fairy prion | Little penguin | Short-tailed shearwater | Fairy prion | Little penguin | Short-tailed shearwater | Fairy prion |
| $\delta^{13}$C (‰) | 2009 | $-19.1 \pm 0.4$ ($n=10$) | $-21.9 \pm 0.4$ ($n=16$) | $-19.2 \pm 0.2$ ($n=8$) | $-19.8 \pm 0.1$ ($n=18$) | $-22.2 \pm 0.3$ ($n=20$) | – | $-19.0 \pm 0.3$ ($n=18$) | $-22.9 \pm 0.5$ ($n=20$) | – |
| | 2010 | $-19.5 \pm 0.2$ ($n=20$) | $-23.1 \pm 0.3$ ($n=20$) | $-19.5 \pm 0.6$ ($n=10$) | $-20.1 \pm 0.4$ ($n=19$) | $-22.2 \pm 0.3$ ($n=20$) | $-20.5 \pm 0.7$ ($n=6$) | $-19.0 \pm 0.2$ ($n=20$) | $-22.1 \pm 0.4$ ($n=20$) | – |
| | 2011 | $-18.9 \pm 0.3$ ($n=20$) | $-21.9 \pm 0.4$ ($n=20$) | $-18.8 \pm 0.4$ ($n=18$) | $-19.9 \pm 0.2$ ($n=10$) | $-22.0 \pm 0.3$ ($n=20$) | $-19.8 \pm 0.3$ ($n=17$) | $-18.5 \pm 0.2$ ($n=10$) | $-21.1 \pm 0.5$ ($n=13$) | – |
| $\delta^{15}$N (‰) | 2009 | $15.5 \pm 0.3$ ($n=10$) | $9.5 \pm 0.6$ ($n=16$) | $13.7 \pm 0.6$ ($n=8$) | $14.4 \pm 0.2$ ($n=18$) | $9.3 \pm 0.7$ ($n=20$) | – | $12.9 \pm 0.9$ ($n=18$) | $9.4 \pm 0.6$ ($n=20$) | – |
| | 2010 | $13.2 \pm 0.9$ ($n=20$) | $8.5 \pm 0.4$ ($n=20$) | $13.6 \pm 0.9$ ($n=10$) | $13.4 \pm 0.5$ ($n=19$) | $8.6 \pm 0.5$ ($n=20$) | $11.7 \pm 0.8$ ($n=6$) | $13.6 \pm 0.3$ ($n=20$) | $8.8 \pm 0.3$ ($n=20$) | – |
| | 2011 | $15.0 \pm 0.4$ $n=(20)$ | $9.7 \pm 0.6$ ($n=20$) | $12.5 \pm 0.9$ ($n=18$) | $14.9 \pm 0.3$ ($n=10$) | $9.9 \pm 0.6$ ($n=20$) | $13.2 \pm 0.5$ ($n=17$) | $13.6 \pm 0.2$ ($n=10$) | $9.9 \pm 0.7$ ($n=13$) | – |

**Table 4  Winter mean (±SD) $\delta^{13}$C and $\delta^{15}$N values (‰) in whole blood of little penguins, fairy prions and common diving petrels from western, central and eastern Bass Strait.**  The samples were collected in winter (July–August) 2008, 2009 and 2010.

| | | Western Bass Strait | | | Central Bass Strait | | | Eastern Bass Strait | | |
|---|---|---|---|---|---|---|---|---|---|---|
| | | Little penguin | Fairy prion | Common diving petrel | Little penguin | Fairy prion | Common diving petrel | Little penguin | Fairy prion | Common diving petrel |
| $\delta^{13}$C (‰) | 2008 | $-19.2 \pm 0.4$ ($n = 3$) | – | – | $-19.0 \pm 0.3$ ($n = 7$) | – | $-20.8 \pm 0.8$ ($n = 10$) | – | – | – |
| | 2009 | $-20.0 \pm 0.4$ ($n = 2$) | – | – | $-19.7 \pm 0.1$ ($n = 20$) | – | $-20.8 \pm 0.2$ ($n = 4$) | $-19.3 \pm 0.5$ ($n = 20$) | – | – |
| | 2010 | $-19.3 \pm 0.6$ ($n = 16$) | $-19.2 \pm 0.4$ ($n = 18$) | $-19.6 \pm 0.1$ ($n = 8$) | $-19.8 \pm 0.1$ ($n = 20$) | $-19.5 \pm 0.3$ ($n = 3$) | $-20.9 \pm 0.3$ ($n = 15$) | $-19.2 \pm 0.3$ ($n = 20$) | – | – |
| $\delta^{15}$N (‰) | 2008 | $13.0 \pm 0.1$ ($n = 3$) | – | – | $14.4 \pm 0.4$ ($n = 7$) | – | $12.0 \pm 1.1$ ($n = 10$) | – | – | – |
| | 2009 | $13.3 \pm 0.4$ ($n = 2$) | – | – | $15.0 \pm 0.4$ ($n = 20$) | – | $14.3 \pm 0.2$ ($n = 4$) | $13.1 \pm 1.2$ ($n = 20$) | – | – |
| | 2010 | $13.2 \pm 0.6$ ($n = 16$) | $11.4 \pm 1.4$ ($n = 18$) | $11.6 \pm 0.3$ ($n = 8$) | $14.2 \pm 0.3$ ($n = 20$) | $12.2 \pm 0.3$ ($n = 3$) | $12.2 \pm 0.2$ ($n = 15$) | $13.3 \pm 0.6$ ($n = 20$) | – | – |

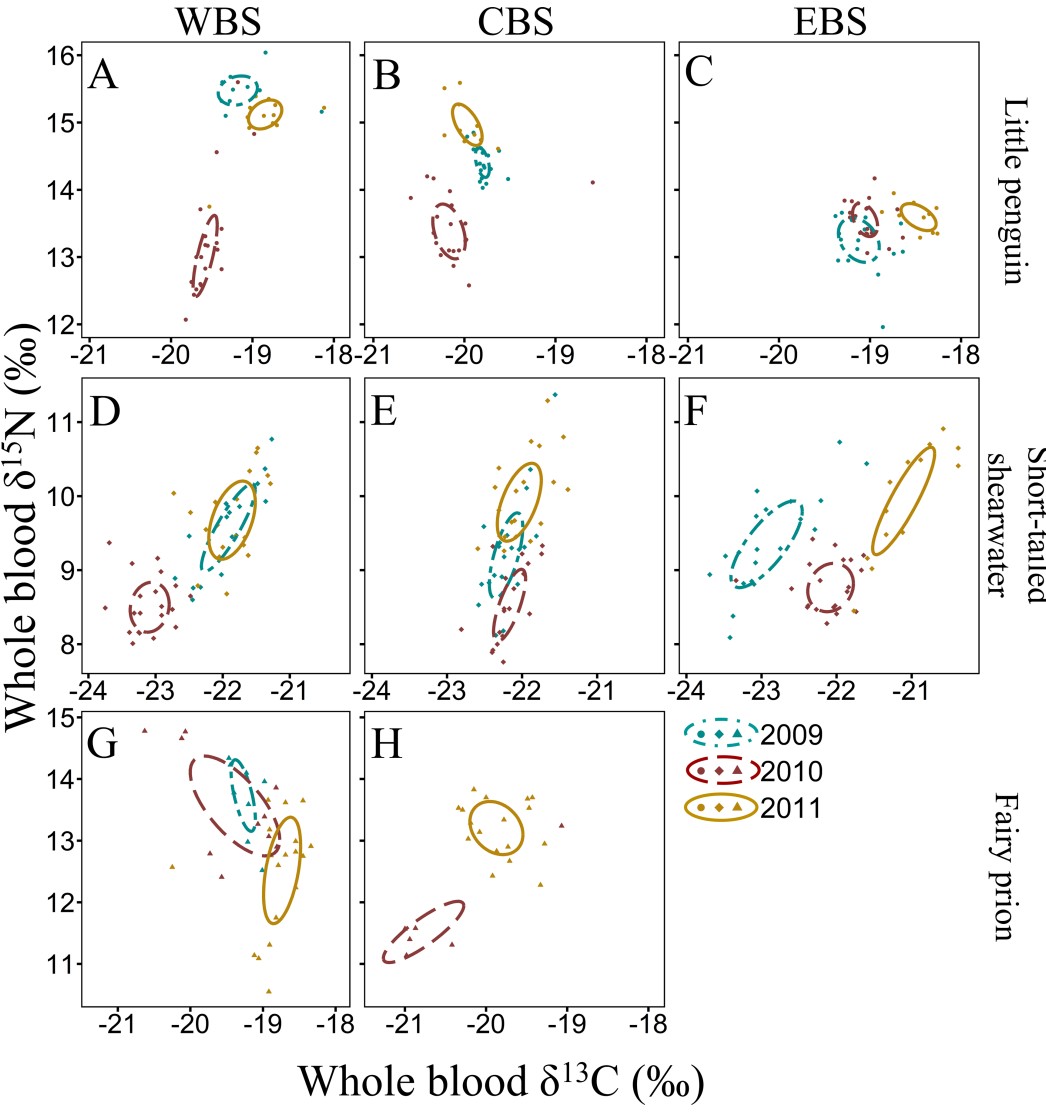

**Figure 3** Summer inter-annual variation of $\delta^{13}$C and $\delta^{15}$N values (‰) in whole blood of little penguins (A, B, C), short-tailed shearwaters (D, E, F) and fairy prions (G, H): western Bass Strait (WBS), central Bass Strait (CBS) and eastern Bass Strait. Solid lines represent the standard ellipses corrected for sample size (SEAc) based on $\delta^{13}$C and $\delta^{15}$N values in summer 2009, 2010 and 2011. Note that the ranges for $x$ and $y$ axes are different for each species.

between region for the little penguin, fairy prion and common diving petrel (Paired $t$-test or Wilcoxon-test: $P < 0.01$). For the short-tailed shearwater, a spatial difference in $\delta^{15}$N values was detected between WBS and EBS in 2010 (Paired $t$-test $t_{37} = -2.19$, $P < 0.05$), but no other differences were found. The models for $\delta^{13}$C retained the region and year as main variables for all the study species, explaining 55% to 70% of the variance (Table S4).

The four study species occupied different isotopic niches in all years and each region (Fig. S1). Values of $\delta^{13}$C and $\delta^{15}$N in whole blood of short-tailed shearwaters were lower (with no isotopic niche overlap) than those of the other species in each region in all three

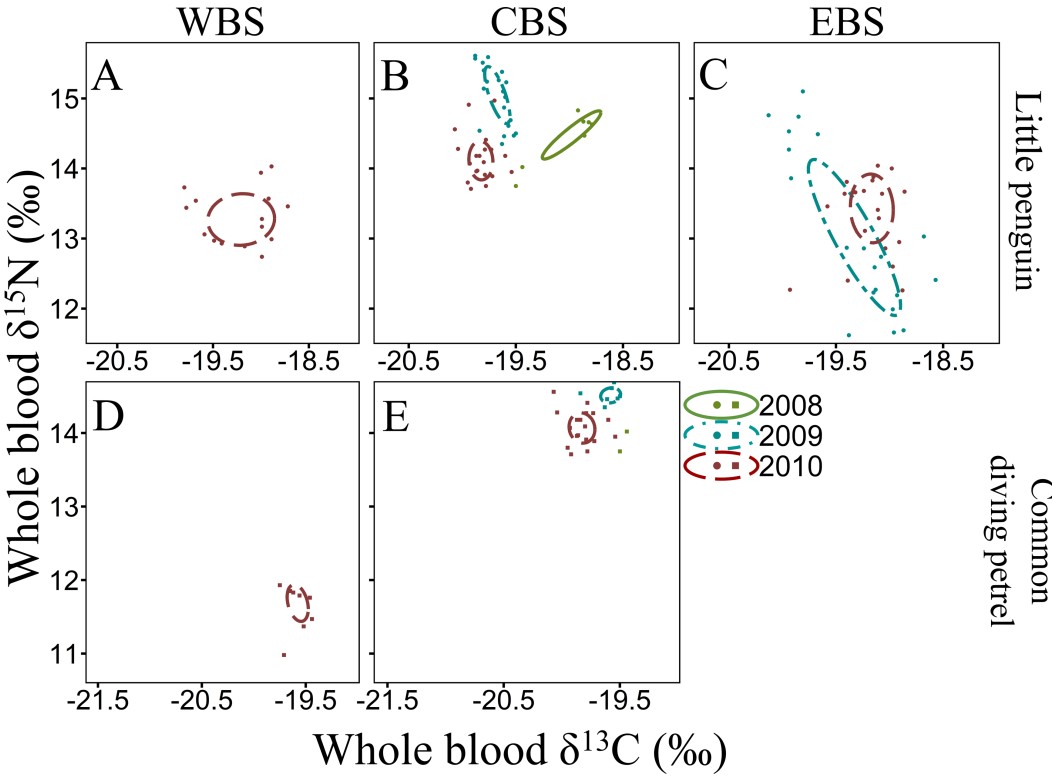

**Figure 4** **Winter inter-annual variation of $\delta^{13}C$ and $\delta^{15}N$ values (‰) in whole blood of little penguins (A, B, C) and common diving petrels (D, E): western Bass Strait (WBS), central Bass Strait (CBS) and eastern Bass Strait (EBS).** Solid lines represent the standard ellipses corrected for sample size (SEAc) based on $\delta^{13}C$ and $\delta^{15}N$ values in winter 2008, 2009 and 2010. Note that the range for $x$ and $y$ axes are different for each species.

years (paired $t$-test or Wilcoxon-test: $p < 0.01$, Table 3 and Table S5). In contrast, mean $\delta^{15}N$ values in blood of little penguins typically showed the greatest enrichment in both winter and summer, though this varied spatially and inter-annually (Tables 3 and 4). Isotopic $\delta^{13}C$ values of little penguins and fairy prions were relatively similar, but tended to be higher than those of common diving-petrels in winter, particularly in CBS. While the isotopic niche of the fairy prion overlapped sparsely with the common diving petrel (SEA$_B$ overlap < 8.1%), and with little penguin in 2009 and 2011 (SEA$_B$ overlap < 4.9%), niche overlap with the little penguin was important in 2010 (maximum SEA$_B$ overlap = 21.8% in winter 2010 in WBS, Table S5).

## DISCUSSION

Foraging niches of sympatric seabird species diverge in a variety of spatial and temporal ways (*Waugh & Weimerskirch, 2003*; *González-Solís, Croxall & Afanasyev, 2008*; *Davies et al., 2009*). Bass Strait is occupied by an abundant marine avifauna (*Ross et al., 2001*), with little previously known of the trophic niches of most species. Combining stomach contents and stable isotope analyses, the present study has shown that the four most abundant and

ubiquitous species generally occupy different trophic niches that vary among regions, years and season.

## Diet

The little penguin is considered an inshore generalist forager relying mostly on small pelagic prey such as Clupeiformes (*Cullen, Montague & Hull, 1992*; *Chiaradia et al., 2010*; *Sutton, Hoskins & Arnould, 2015*). In the present study, stomach contents of little penguins in summer were similar to that previously reported, with Australian anchovy and post-larval fish contributing the majority of samples. In contrast, winter stomach contents were dominated by jack mackerel, highlighting a seasonal switch in the availability of the main prey of little penguins. Such differences have also been shown in little penguins from Albatross Island in southern Bass Strait (*Gales & Pemberton, 1990*) and Phillip Island in northern Bass Strait (*Cavallo et al., 2018*). While recent studies have observed that jellyfish can contribute a substantial proportion of the little penguin diet (*Sutton, Hoskins & Arnould, 2015*; *Cavallo et al., 2018*), no evidence of such prey were found in the present study. This could potentially be due to rapid digestion of gelatinous prey in comparison to fish or crustaceans (*Cavallo et al., 2018*), emphasising the limitation of traditional stomach content analyses, or reflect inter-annual differences in available prey types.

Stomach contents of short-tailed shearwaters in the present study were similar to those of individuals from Tasmania (*Weimerskirch & Cherel, 1998*; *Cherel, Hobson & Weimerskirch, 2005*). The main identified prey was the coastal krill, indicating that sampled birds were mostly returning from short foraging trips over the continental shelf (*Blackburn, 1980*; *Weimerskirch & Cherel, 1998*). Indeed, during the breeding season, short-tailed shearwaters alternate between short (1–2 d) local trips within 35–70 km of the colony and long trips (10–20 d) to Antarctic waters (*Weimerskirch & Cherel, 1998*; *Woehler, Raymond & Watts, 2006*; *Raymond et al., 2010*; *Einoder et al., 2011*; *Berlincourt & Arnould, 2015b*) where they feed mainly on coastal krill, and myctophid fish and Antarctic krill, respectively (*Montague, Cullen & Fitzherbert, 1986*; *Weimerskirch & Cherel, 1998*). In the 2010 samples, a limited number of birds ($n = 4$) had stomach contents dominated by stomach oil and digested *Euphausia* sp, suggesting they had returned from long trips to Antarctic waters (*Weimerskirch & Cherel, 1998*).

Coastal krill occurs in neritic waters of eastern Australia, where other krill species are rare or absent (*Blackburn, 1980*). Due to its abundance, it plays a key role in the coastal ecosystem, reflected by its dominance in the diets of various cetacean, seabirds and fish species (*O'Brien, 1988*; *Gill et al., 2011*; *Woehler et al., 2014*). Despite limited data on the diet of fairy prions and common diving petrels in Bass Strait, their stomach contents confirmed the importance of coastal krill to these species in the Australasian region (*Harper, 1976*; *Schumann, Arnould & Dann, 2008*). These results, together with estimates of trip duration in previous studies (1–3 d trips at sea, *Harper, 1976*; *Bocher Cherel & Hobson, 2000*; *Bocher, Labidoire & Cherel, 2000*; *Navarro et al., 2013*; *Zhang et al., 2018*), suggest that both breeding fairy prions and common diving petrels most likely forage within Bass Strait or in the vicinity of the continental shelf. This analysis emphasises the value of a multi-tools approach when considering niche segregation, as here, while stomach

analysis can suggest substantial dietary overlap among the procellariiforms, isotopic and tracking analysis may be able to tease the species apart into separate foraging niches.

## Spatial variability in isotopic niche

Since whole blood integrates dietary information over approximately four weeks (*Bearhop et al., 2002*), it might be expected that isotopic values for short-tailed shearwaters would reflect a combination of both their local and Antarctic foraging areas (*Berlincourt & Arnould, 2015b*), thereby masking any differences in blood isotope values between foraging zones. However, *Cherel, Hobson & Weimerskirch (2005)* showed that while most of the food consumed by short-tailed shearwaters during short local trips is allocated to their chick, adults feed for themselves when foraging farther south and, therefore, have a truly Antarctic blood isotopic signature. In the present study, values of $\delta^{13}$C in whole blood of short-tailed shearwaters were comparable to those of other procellariiforms foraging in subantarctic waters (*Cherel et al., 2002a*; *Cherel et al., 2002b*; *Quillfeldt, McGill & Furness, 2005*) but were slightly higher than those in plasma of short-tailed shearwaters from south Tasmania (*Cherel, Hobson & Weimerskirch, 2005*).

Interestingly, the $\delta^{13}$C values of short-tailed shearwaters varied significantly between the three study regions in Bass Strait, indicating possible foraging spatial segregation within the Southern Ocean by individuals from the different regions. As $\delta^{13}$C values are higher in subtropical than in Antarctic waters (*Cherel & Hobson, 2007*; *Jaeger et al., 2010*), this suggests a latitudinal segregation in the foraging areas between the three sampled populations. However, this variation was not consistent across years, with strong inter-annual variability in $\delta^{13}$C values for birds from WBS and EBS compared to CBS. This could suggest not only colony-specific niche segregation but also density-dependent competition (*Ainley et al., 2004*; *Wakefield et al., 2013*), with the smallest colonies having a more flexible foraging area. Indeed, the CBS population is considerably bigger than the WBS and EBS populations, with 755,400, 30,000 and 6,000 breeding pairs, respectively (*Bowker, 1980*; *Pescott, 1976*; *Fullagar & Heyligers, 1996*; *Schumann, Dann & Arnould, 2014*). This is in accordance with previous tracking studies (*Berlincourt & Arnould, 2015b*) that observed inter-annual longitudinal and latitudinal variation in the long trip foraging areas of short-tailed shearwater from the small populations in WBS and EBS. Despite the geographic and temporal differences in $\delta^{13}$C observed, there were no major differences in the $\delta^{15}$N values between regions in the present study, highlighting the consistency of the diet of short-tailed shearwaters in the Southern Ocean.

For little penguins, fairy prions and common diving petrels, seasonal and geographic differences in isotopic signatures are likely to reflect differences in prey availability associated with the strength of the prevailing of ocean currents and upwelling systems in the different regions of Bass Strait. For example, the SAC may transport cold waters from the west into Bass Strait (*Mickelson, Dann & Cullen, 1992*; *Sandery & Kämpf, 2007*), weakening towards the east (*Sandery & Kämpf, 2007*) where the EAC increases in prevalence, bringing warmer nutrient-poor water into north-eastern Bass Strait (*Gibbs, 1992*). This was reflected in $\delta^{13}$C values in the whole blood of little penguins and fairy prions, where values were systematically lower in CBS than in WBS and EBS. Similar observations have been reported

for Australian fur seals where $\delta^{13}$C values in blood plasma of individuals from CBS were consistently lower than those from EBS (*Arnould et al., 2011*). In winter, however, spatial differences in isotopic values declined. This may reflect the homogenization of Bass Strait waters in winter due to a greater influence of the SAC and SASW during this period (*Prince, 2001*; *Sandery & Kämpf, 2007*).

## Inter-annual trophic variability

The isotope values in the whole blood of little penguins from WBS and CBS in summer are within the range previously reported from Phillip Island in CBS (*Chiaradia et al., 2010*; *Chiaradia et al., 2012*), with the exception of 2010 when $\delta^{15}$N values were significantly lower in both regions. This could reflect fluctuations in isotopic baseline signatures due to different water masses and variable strength of the currents influencing the regions where individuals foraged. Indeed, as reported by *Polito et al. (2019)*, variations in oceanic factors such as chlorophyll-*a* concentration can substantially alter mean isotope values independently of any change in the diet of the species. However, inter-annual variation in $\delta^{15}$N values could also reflect a variation in main prey species consumed, with little penguins known to have important inter-annual variability in their diet (*Gales & Pemberton, 1990*; *Cullen, Montague & Hull, 1992*; *Chiaradia et al., 2010*). In the present study, individuals from CBS in summer 2009 consumed predominantly Australian anchovy, a species exploiting higher trophic levels than other prey targeted by little penguins (*Espinoza et al., 2009*; *Van der Lingen et al., 2009*). As previously highlighted by *Chiaradia et al. (2010)*, $\delta^{15}$N values are higher during years with an important proportion of anchovy in little penguin diet. Therefore, it is likely that the low $\delta^{15}$N values in summer 2010 in the present study was due to a depletion of Australian anchovy in the diet, potentially due to a reduced availability in the region. This is consistent with previous reports indicating inter-annual flexibility in little penguin at-sea foraging behaviour in relation to environmental conditions that directly influence prey abundance (*Berlincourt & Arnould, 2015a*; *Camprasse et al., 2017*). Indeed, in WBS and CBS, the niche space occupied by little penguins in 2010 was much larger than in 2009 and 2011, indicating a larger trophic diversity (*Layman et al., 2007*), possibly due to the absence of the usual main prey. Similarly, for both fairy prions and common diving petrels, $\delta^{15}$N values in the whole blood of both species in the region varied substantially between years, suggesting a potential variation in the importance of their main prey (coastal krill) in their respective diet. Significant inter-annual differences in the density and biomass of coastal krill in southern Bass Strait have previously been observed (*Young et al., 1993*).

## Trophic and isotopic niche segregations

In the present study, interspecific comparisons of stomach contents and $\delta^{15}$N values revealed that little penguins typically occupied the highest trophic positions of the four seabird species while short-tailed shearwaters always occupied the lowest. Little penguin $\delta^{15}$N values were nevertheless lower than those of the top predators shy albatross and Australian fur seal (*Arnould et al., 2011*; *Cherel et al., 2013*), but were close to the values of the large Australasian gannet (*Angel, Berlincourt & Arnould, 2016*), that predominantly

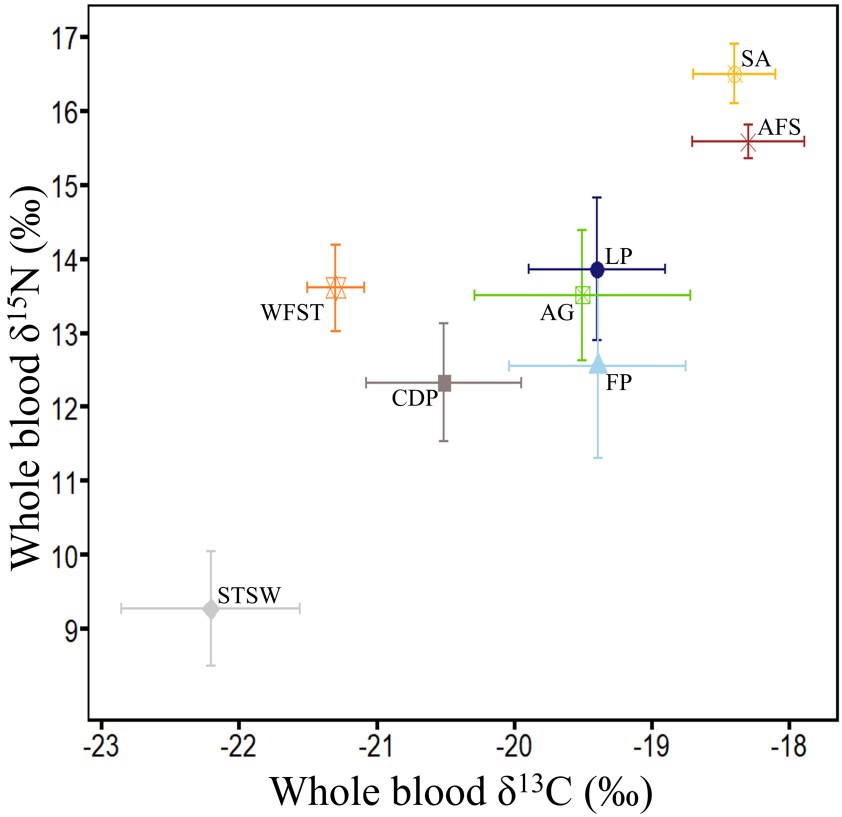

**Figure 5** **Summary of $\delta^{13}C$ and $\delta^{15}N$ values (‰) in whole blood of the main marine predators in Bass Strait region.** Little penguin (LP, $n = 278$; present study; summer and winter combined), short-tailed shearwater (STSW, $n = 177$; present study; summer), fairy prion (FP, $n = 88$; present study; summer and winter combined), common diving petrel (CDP, $n = 38$; present study; winter), Australasian gannet (GA, $n = 27$; *Angel, Berlincourt & Arnould, 2016*; summer), white-faced storm petrel (WFST, $n = 17$; *Underwood, 2012*; summer), shy albatross (SA, $n = 8$; *Cherel et al., 2013*; summer) and Australian fur seal (AFS, $n = 242$; *Arnould et al., 2011*; winter). The isotopic values of WFST and SA were calculated from data on chick feathers (*Underwood, 2012*) and adult feathers (*Cherel et al., 2013*), respectively, and corrected using mean correction factors in *Cherel et al. (2014)*.

consumes pilchards and anchovy (*Bunce, 2001*) (Fig. 5). In contrast, $\delta^{15}N$ values of short-tailed shearwaters, fairy prions and common diving petrels were remarkably lower than those of the much smaller white-faced storm petrels (*Underwood, 2012*) (Fig. 5), which consume a significant proportion of fish in addition to coastal krill (*Underwood, 2012*). These results, combined with the stomach content analysis, confirm that coastal krill was a key prey taxon in all three procellariforms in central Bass Strait. During breeding, both fairy prions and common diving petrels return to the nest every night (*Harper, 1976*; *Payne & Prince, 1979*), suggesting that they forage mainly on the shelf near their colonies. Elsewhere, fairy prions take prey from the surface waters (*Harper, 1987*; *Prince & Morgan, 1987*) whereas common diving petrels exploit depths averaging 2–4 m (*Navarro et al., 2013*; *Navarro, Votier & Phillips, 2014*; *Dunphy et al., 2015*). Likewise, despite isotopic signatures showing self-maintenance feeding in the Southern Ocean,

short-tailed shearwaters forage on coastal krill over the shelf near colonies during short trips (*Einoder et al., 2011*; *Berlincourt & Arnould, 2015b*), resulting in the potential for interspecific overlap in the foraging zones of the three procellariforms. However, short-tailed shearwaters forage at deeper depths (average 13 m) during local trips (*Weimerskirch & Cherel, 1998*). While the foraging zones and dive depths of little penguins may overlap with those of short-tailed shearwaters (*Berlincourt & Arnould, 2015a*; *Berlincourt & Arnould, 2015b*), the limited distance travelled per trip and fish-based diet of little penguins would reduce competition with procellariforms.

In addition to segregation of diet and foraging behaviour, the four species differ in their breeding phenologies. Common diving petrels, fairy prions and short-tailed shearwaters lay their eggs in late July, late October and late November, respectively (*Harris, 1979*; *Marchant & Higgins, 1990*). Thus, there is limited overlap in the breeding periods of the three procellariforms. However, the protracted and variable breeding season of little penguins (*Reilly & Cullen, 1981*; *Cullen, Montague & Hull, 1992*) may overlap with the other species. Interspecific competition may, therefore, intensify in years of low fish prey availability. Indeed, previous studies have documented the presence of coastal krill in the diet of little penguin during years of low prey availability (*Cullen, Montague & Hull, 1992*).

The dependence of these seabird species on relatively few prey types (such as coastal krill, pilchard or anchovy) may increase the impacts of reductions in prey abundance. Climate models have described an intensification of the EAC due to large-scale changes in ocean circulation in the Southern Hemisphere (*Cai, 2006*), produced in association with an increasing trend in the Southern Annular Mode (*Cai et al., 2005*). During years of intensified EAC, *Young et al. (1993)* reported a dramatic drop in coastal krill biomass. This is likely to adversely affect seabirds in the region (*Mills et al., 2008*; *Chambers et al., 2011*). Similarly, significant mortality events, poor chick growth and population declines in short-tailed shearwaters in Tasmania have been previously attributed to local prey shortages (*Vertigan, 2010*). Declines in coastal krill availability may also indirectly impact little penguins since this species is an important dietary component of several of their prey taxa (*Harris et al., 1991*; *O'Brien, 1988*). The predicted increase in the strength of the EAC with climate change (*Cai et al., 2005*) could, therefore, have severe negative consequences for the Bass Strait seabird community (*Chambers et al., 2011*).

## CONCLUSIONS

In summary, the present study has shown that the isotopic niches of seabirds in Bass Strait vary significantly between regions, years and seasons. These differences are likely to result from changes in prey availability driven by variations in ocean currents and local productivity. Despite interspecific similarities in diet, divergence in the relative foraging niche is likely to reduce interspecific competition for prey, though this may become more important in years of low prey availability. The low diversity of prey taxa ingested by these seabirds suggests that they are vulnerable to changes in the availability of key prey. In order to better understand the foraging niches of the Bass Strait community of seabirds, as well as their capacity to adapt to changing environmental conditions, more

detailed information on their foraging zones and feeding behaviour is required. This is particularly important for the small procellariforms in light of the paucity of information in south-eastern Australia, in contrast to the numerous studies that have been conducted on little penguins and short-tailed shearwaters in the region (e.g., *Ropert-Coudert et al., 2004*; *Ropert-Coudert, Kato & Chiaradia, 2009*; *Cleeland, Lea & Hindell, 2014*; *Berlincourt & Arnould, 2015a*; *Berlincourt & Arnould, 2015b*). Such information may help elucidate the likelihood of interspecific competition in this assemblage of seabirds.

## ACKNOWLEDGEMENTS

The contribution of Dr Roger Kirkwood and Dr Michael Lynch who collected blood samples from Lady Julia Percy Island during some seasons, and Dr Tom Montague who provided information on identification of fish prey remains using mouth parts and scales is gratefully acknowledged. We thank Dr Alice Carravieri who provided useful comments on the later version of the manuscript, and Dr Tim Poupart and Dr Paul Tixier for their respective help on statistical analysis and R code.

### Funding
Funding was provided by the Winifred Violet Scott Charitable Trust fund and Stuart Leslie Bird Research Award. The funders had no role in study design, data collection and analysis, decision to publish, or preparation of the manuscript.

### Grant Disclosures
The following grant information was disclosed by the authors:
Winifred Violet Scott Charitable Trust fund and Stuart Leslie Bird Research Award.

### Competing Interests
The authors declare there are no competing interests and Peter Dann is employed by Phillip Island Nature Parks.

### Author Contributions
- Aymeric Fromant analyzed the data, prepared figures and/or tables, authored or reviewed drafts of the paper, and approved the final draft.
- Nicole Schumann conceived and designed the experiments, performed the experiments, authored or reviewed drafts of the paper, and approved the final draft.
- Peter Dann and John P.Y. Arnould conceived and designed the experiments, authored or reviewed drafts of the paper, and approved the final draft.
- Yves Cherel analyzed the data, authored or reviewed drafts of the paper, and approved the final draft.

### Animal Ethics
The following information was supplied relating to ethical approvals (i.e., approving body and any reference numbers):
All research was conducted under permit from Deakin University (AWC A9-2008).

## Field Study Permissions

The following information was supplied relating to field study approvals (i.e., approving body and any reference numbers):

Field experiments were approved by the Department of Sustainability and Environment (Permit No. 10004530), and access to the islands was provided by Parks Victoria.

## Data Availability

The raw data are available in a Supplementary File.

## Supplemental Information

Supplemental information for this article can be found online at http://dx.doi.org/10.7717/peerj.8700#supplemental-information.

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
