# Peer review of "Trophic niches of a seabird assemblage in Bass Strait, south-eastern Australia"

_PeerJ, doi:10.7717/peerj.8700_

## Round 0.1 · original submission · Major Revisions

Dear Dr Fromant and co-authors,

I have received three detailed reviews from experts in the field.
They have provide comments, queries and suggestion that we request you provide specific and detailed responses to.

In particular I would like to draw your attention to several aspects:

References

References need to be brought up to date, and it is suggested you reference research in areas other than Phillip Island, such as the rest of Victoria, Tasmania and internationally

Statistics

If the same birds were sampled in different seasons then please add this as a random factor in your model, either way please clarify. Also, please consider investigating season as a random factor to ascertain whether isotope signature changes between seasons.

Graphics/graphs

More background on the breeding cycle, and which sections your data refers to, are required, perhaps in a diagram.

Please consider Tables 3 and 4 as figures to simplify the message.

We look forward to your revised manuscript shortly.

Regards

Steve

Reviewer 1 ·

Basic reporting

Literature use needs to be improved - see comments below. Many key refs missing, too many old citations, need to include studies from beyond Phillip Island.

Experimental design

No comment

Validity of the findings

No comment

Additional comments

Referee’s comments – Fromant et al seabirds Bass St


Line 38. Surely more recent and more relevant refs than Ridoux 1994?
Lines 37 -41 – would like to see more references than 1 per aspect, and more recent refs where possible.
Line 51. Worth noting that 14.6 of 15M seabirds are shearwaters – diversity is actually low given predominance of one species.
Table 1 – prob worth including the dates of the population estimates – some of the Brothers et al surveys date back to the 1970s, 1980s and 1990s – well out of date.
Lines 60 – 64. The SASW is primarily present in the winter months when the STC moves north into Bass St or even farther north.
Line 84. Surely cite the recent surveys of Schuman rather than the national synthesis of Ross 2001?
Lines 86 – 90 – the foraging range of short-tailed shearwaters extends to the Antarctic in the summer breeding season – clearly worth citing these studies.
Line 94 – if there is no information, then no references are needed, but if there are some studies to generate “almost no” information – please cite. Text should be at line 106, not 113 – 124.
Line 106 – how were stomach samples obtained? Method and references pls. Collected at sea?
Line 140 – presumably squid beaks?
Line 214 – what is “coastal krill”? Nyctiphanes? Please provide latin names where possible in the text [I don’t have the supp material]
Line 239. Can the low variance explained by the foraging areas of the shearwaters?
Line 292 – shearwaters feed on superba – Antarctic Krill, not “subantarctic” krill.
Lines 303/304. Presumably we are talking about breeding birds foraging over Bass St/shelf. Prions and diving petrels will travel well into the Southern Ocean during non-breeding season.
Line 313 – ‘farther’ south!
Lines 320-323 – could the differences be related to shearwaters flying west from WBS colonies and east from EBS colonies? Local/fine-scale variation close to colonies?
Lines 326 – 329 – probably worth thinking about some of Ainley’s work re colony size/foraging ranges/overlaps here. Dare we suggest Ashmole’s Halo?
Line 338 – what evidence is there that the SAC water contains prey and nutrients? Any independent data to support this claim?
Line 347 – does the SAC provide the greatest volume of water into the study area in winter? What about the northward movement of the STC?
Line 387 – ‘coastal krill’ – Nyctiphanes? If so, worth looking at the role of Nyctiphanes in New Zealand (Mills et al 2008) and in SE Tasmania (Woehler et al 2014) re gulls and other seabirds.
Lines 409+ - some discussion re potential impacts of climate change on Australian seabirds has been reviewed by Chambers et al and should be included/cited in Discussion here.
Line 417 – Did Vertigan report on Bass St shearwaters?

Reviewer 2 ·

Basic reporting

No comment

Experimental design

No comment

Validity of the findings

No comment

Additional comments

This study compares isotopic values of blood from three abundant seabird species over two seasons (summer & winter) at several locations across Bass Strait to identify niche overlap between these species. It is very well written, and tables and figures are clear, although in the case of the figures I recommend the use of line styles/symbols that make it possible to distinguish between lines when copied in B&W. My comments are minor.
Methods
Lines 137 – 140. How were eroded otoliths dealt with? What proportion were still identifiable to species?
Explain why discrimination factors weren’t used for isotopic analyses.
Statistical analyses: were the same individuals sampled twice across seasons? If so, individual would have to be entered as a random factor into models.
Line 211. Re-organise this sentence so that the crustacean remains are not consuming the jack mackerel.
Lines 225 and on: Report results in past tense.
Lines 237-238: These differences are not very obvious as they are rather small.
Line 239: substitute “but” for ‘and’. I found this paragraph hard to follow.
Lines 241-242. I would prefer to see Table S4 in the main body of text, but reduced, showing only models within delta AIC values of 2. This I believe is the commonly adopted method for presenting these models.
Line 251: I found this figure really helpful and suggest it should be in the main body of text rather than in supplementary material.
Line 254: add ‘in’ before ‘all’.
Discussion
Lines 282-283: The point of the Cavallo et al. paper is that gelatinous prey is virtually undetectable without doing DNA analyses.
Line 305: At the end of this section I suggest making a statement on how conventional stomach analysis suggests substantial dietary overlap among the procellariforms, and that the isotopic analysis is able to tease apart into separate foraging niches, to emphasise the value of this dual approach.
Line 358. Specify what “this’ is.
Line 373: Please clarify how variation in the consumption of coast krill would result in substantial inter-annual variation in N15 values.
Line 381: Substitute “than” for “to”.
Line 391: Does this difference in foraging strategy explain the difference in C13 values between FP CDP? If not, what might?

Reviewer 3 ·

Basic reporting

Background/Context:
To better understand your results, the reader needs much more background on the breeding cycle of the species studied and on which part of the breeding cycle your data refer to. Because seabirds are central place foragers, this information is extremely important. It could be helpful to create a figure that shows the annual cycle of each species with the period your sampling covers highlighted somehow.

Figures and Tables:
As is, tables 3 and 4 are a lot to take in. Maybe you could simplify them to make it easier for the reader to digest the key information? For example, the values for the mean and range of each isotope are better displayed as a figure (i.e., as in Figure 4). For the Figure 4 legend, please clarify whether these are data aggregated across summer and winter.

Experimental design

Animal Use Permit:
It is unclear to me whether an animal use permit was obtained for the study. Maybe this was the permit from Deakin University? Please clarify which permit process thoroughly reviewed your animal use procedure, as the methods used are fairly high impact (especially the stomach flushing).

GLMs:
It would be interesting to know whether season was a significant predictor of isotope value. Could you test models including season as a term, rather than separating out the two seasons a priori and running separate models?

Validity of the findings

Caveats
Although you do not need to go into too much detail, please provide readers unfamiliar with stable isotopes with some of the caveats of this approach. For example, your results could have been influenced by difference in TEFs among species. Furthermore, the isotope data represent diet aggregated over time, while the stomach flushing data represent a snapshot of diet.

Additional comments

Your manuscript is well written and provides useful information on the trophic niche of the four species studied. The raw data are provided in appropriate detail in the supplementary files (though they could be organized a bit better for ease of examination by reviewers/readers) and the statistical approach is appropriate. There are, however, several places where more details are necessary for the reader to interpret your findings. Furthermore, several changes could be made to improve readability.

-Lines 52-53 and elsewhere in the document, the word "which" is used where the word "that" should be used.
-Line 90 – If these references are species specific, please move them up in the sentence to follow the relevant species.
-Line 101 – Were data collected throughout this time range in each season? As noted above, please provide more context for the portion of the breeding cycle that these isotope values represent.
-Line 126 – Why weren't meals provided to the other species studied?
-Line 172 – Please provide reference for why C:N values of 3.1-3.7 do not require lipid extraction.
-Lines 210-213 – This sentence should be reworded to improve readability.
-Lines 226-231 – Do these range values include both summer and winter data? In general, summer and winter data are combined in some portions of the paper and not others (e.g., for the GLMs) and it is unclear when/why the decision to combine or not combine data was made.
-Lines 251-261 – Please provide a supplemental table showing niche overlap among all pairs of categories (each species in each season x each species in each season). You should also define niche overlap for those not familiar with SIBER – I am assuming you used: area of overlap/combined area of niches.
-Lines 251-261 – Why use fairy penguin as the species to which all other species are compared in table 3? Based on previous studies, isn't it expected that overlap between fairy penguins and the other species studied will be low (e.g., as outlined in Lines 84-90)? It might be more interesting to compare the species that feed primarily on krill to each other.
-Lines 264-265 – Please reword this sentence.
-Lines 282-284 – Is there anything from the SIA results that could be used to indicate the consumption of jellyfish?
-Line 307 – This transition feels odd because it is a continuation of the previous section where the diet of short-tailed shearwaters is discussed. Maybe the discussion could be rearranged slightly to improve flow.
-Line 371 – Please start a new paragraph and clarify what is meant by "similarly" in this sentence. What are these species consuming in years when krill abundance is low?
-Lines 380-383 – This sentence should be reworded to improve readability.
-Lines 385 – Clarify here that the data for white-faced storm petrels are from previous studies (and I am assuming, a different year? Possibly a different region?) i.e., "lower than values previously reported for white-faced storm petrels foraging in this region".
-Lines 386-388 – It's not clear that the preceding sentence indicates that krill are an important diet item for the procellariforms you studied. You could use your stomach content data to back this point up.
-Lines 386-388 – Are there any data availability for the isotope values of prey items I this region? And how variable these signatures are?
-Lines 409-421 – The transition to this paragraph feels somewhat abrupt. Maybe it should be separated out into its own section or better integrated into the rest of the discussion?

---

## Round 0.2 · Minor Revisions

Dear Dr Fromant and co-authors,

Congratulations on a thorough and detailed response to the comments and suggestions of the three expert reviewers.

One of these reviewers has provided key recommendations in response, and I invite you to undertake further minor revisions to refine the manuscript.

I strongly recommend that you provide the additional data to allow the readers to interpret the post-hoc tests fully.
Similarly, it seems to me a reasonable suggestion to restructure the Discussion around the three objectives. This will lead to a far more satisfying and informative read for your target audience.
We look forward to a revised manuscript in the coming weeks.

Regards
Steve

Reviewer 3 ·

Basic reporting

See comments to author

Experimental design

See comments to author

Validity of the findings

See comments to author

Additional comments

Fromant and colleagues' revisions improved the manuscript, particularly in terms of providing important methodological details that were previously missing. However, there are still a several details missing from the results section and the discussion remains difficult to follow (see below).

Lines 16-17: This sentence could be restructured to improve readability. I am also unsure whether "foraging niche" is the appropriate term for this study, which focused on the isotopic niche or dietary niche. Foraging niche is often studied using data on foraging locations (e.g., tags). Isotopes provide only general insight into foraging location.
Line 22: Since you didn't sample all species in all sites/seasons years, maybe instead say "that varied among years, seasons, and regions"
Line 94: inter-annual, seasonal, and geographic variation
Line 166: Please clarify "(not sampled for diet determination)"- does this just mean that none of the individuals were sampled for both gut contents and isotopes?
Line 212: Were the stomachs without identifiable remains empty, or were the items just too digested to identify? (especially for the common diving petrels)
Line 238 and elsewhere: I know a lot of tests were run, and commend the authors for providing the raw data, but more details are needed on the results from the post-hoc tests. There is now wide agreement in the literature that p values should not be included in papers without additional information necessary for the reader to reach their own conclusions regarding the results. For example, see recent papers by Smith "Ending Reliance on Statistical Significance Will Improve Environmental Inference and Communication" Estuaries and Coasts https://doi.org/10.1007/s12237-019-00679-y. Furthermore, please provide the actual p-value when you can round it to only two decimal places (e.g. 0.71 instead of >0.70).

Line 245: It is unclear to me why δ13C was included in the model. Is the idea that δ13C might indicate different foraging locations, which would in turn influence the trophic level the individual was feeding at? Some relationship between δ13C and δ15N is often expected due to slight trophic fractionation of δ13C, but I don't see why that would be a meaningful thing to test for in the current study.

Line 249: What is the 0.2%? Variance explained by season for little penguin?

Discussion: I am still having trouble with the structure of the discussion. As noted by the authors, part of this has to do with the complicated nature of the data, especially given that different combinations of samples (isotopes, gut content) and sampling occasions (region, year, season) were collected for each species. I am not sure of the best way to simplify this to make it easier for the reader to follow. One issue is that some sections focus in detail on one species then only cover the other three species very briefly (e.g., inter-annual trophic variability section focuses almost exclusively on little penguins, with just 1-2 sentences on the other species). Maybe restructure the discussion with the three goals outlined in the introduction as the subheadings?

Lines 442-444: This seems like the major finding of the study and could possibly be used to think about restructuring some of the discussion.

---

## Round 0.3 · accepted · Accept

Dear Fromant and co-authors, thank you for taking the time to align the Discussion with revised objective statements, and for dealing with many of the other suggestions of R3. I am happy to recommend your manuscript be accepted.

Regards
Steve